# Imported severe malaria and risk factors for intensive care: A single-centre retrospective analysis

**Alessandra D'Abramo**[1]☯, **Luciana Lepore**[1]☯, **Marco Iannetta**[1]*, **Saba Gebremeskel Tekle**[1], **Angela Corpolongo**[1], **Laura Scorzolini**[1], **Nazario Bevilacqua**[1], **Andrea Mariano**[1], **Maria Letizia Giancola**[1], **Antonella Vulcano**[1], **Micaela Maritti**[1], **Alessandro Agresta**[1], **Mario Antonini**[1], **Umberto D'Alessandro**[2], **Emanuele Nicastri**[1], **Spallanzani Group for Malaria Study**¶

**1** National Institute for Infectious Diseases, IRCCS, Lazzaro Spallanzani, Rome, Italy, **2** MRC Unit The Gambia at the London School of Hygiene and Tropical Medicine, Atlantic Boulevard, Fajara, The Gambia

☯ These authors contributed equally to this work.
¶ Membership of the Spallanzani Group for Malaria Study is provided in the Acknowledgments
* marco.iannetta@inmi.it

**Data Availability Statement:** All relevant data are within the manuscript and its Supporting Information file.

## Abstract

### Objectives

This study aims to identify the risk factors for intensive care (IC) in severe malaria patients admitted to the "Lazzaro Spallanzani" National Institute for Infectious Diseases, Rome, Italy.

### Methods

All patients with confirmed severe malaria and hospitalized between 2007 and 2015 were included in the analysis and stratified into two groups: those requiring IC and those who did not. Five prognostic malaria scores were estimated; clinical severity at IC unit admission was assessed using the Sequential Organ Failure Assessment and the quick-Sequential Organ Failure Assessment scores. Univariate and multivariate analysis were performed to assess factors independently associated to IC.

### Results

A total of 98 severe malaria patients were included; 10 of them required IC. There were no deaths or sequelae. Patients requiring IC had higher severity scores. At the multivariate analysis, only the number of World Health Organization criteria and the aspartate amino-transferase value were independently associated with the need of IC.

### Conclusions

An early and accurate assessment of the severity score is essential for the management of severe malaria patients.

**Funding:** This study was supported by Ricerca Corrente and Ricerca finalizzata WFR PE-2013-02357936 funded by the Italian Ministry of Health to EN.

**Competing interests:** The authors have declared that no competing interests exist.

## Introduction

Imported malaria is a substantial clinical and epidemiological problem in many European countries currently considered as malaria-free. In Europe, approximately 6000 imported malaria cases are reported annually, with 10% of them progressing towards severe malaria [1]. The case-fatality rate of adequately treated severe malaria, due to *Plasmodium falciparum* is about 10–20% while it is almost 100% if left untreated [2]; among patients admitted to intensive care unit (ICU), such case-fatality rate ranges between 5 to 10% [3]. Individuals from non-endemic countries travelling to high malaria endemic countries are immunologically naïve as they have not been previously exposed to the infection. For this reason, when infected, they may evolve rapidly towards severe malaria with high parasite densities, multi-organ involvement, and eventually death [4–6]. Therefore, identifying among these patients those with a higher risk of adverse outcome may improve their management and thus reduce the risk of death. The aim of this study was to assess the risk factors associated with the need of IC. For this purpose, the clinical records of a cohort of severe malaria patients admitted over 8 years at the "Lazzaro Spallanzani" National Institute for Infectious Diseases (INMI), Rome, Italy were analysed.

## Methods

In this single-centre cross sectional study, we retrospectively recruited a total number of 98 patients consecutively hospitalized with confirmed severe malaria at the INMI Spallanzani, Rome, Italy, from January 2007 to December 2015. The INMI Spallanzani is a 200-bed hospital dealing mainly with infectious and tropical diseases with a 10-bed ICU. This study was approved by the Ethic Committee of the INMI Spallanzani (ethics number 38/2016). Inclusion criteria: age >18 years old, written informed consent at hospital admission from the patient or next of kin if patient unable, severe confirmed malaria diagnosis. Severe malaria was diagnosed in presence of *P. falciparum* asexual parasitaemia and at least one clinical (impaired consciousness or unarousable coma; prostration; failure to feed; multiple convulsions–more than two episodes in 24 hours; respiratory distress; circulatory collapse or shock; clinical jaundice and evidence of other vital organ dysfunctions; haemoglobinuria; abnormal spontaneous bleeding; pulmonary oedema) or laboratory (hypoglycaemia; metabolic acidosis; severe anaemia; hyperparasitaemia; renal impairment hyperlactataemia) criterion in absence of any other obvious cause [7]. For *Plasmodium vivax*, all the criteria were applied with the only exception of hyperparasitaemia [7]. Demographic characteristics, medical and travel history, clinical presentation, anti-malarial and supportive treatment, parasitaemia before and during treatment, complications during treatment, adverse drug reactions, clinical outcome (survival, death or sequelae) at day 28 post-treatment were collected for all patients from the clinical record. In addition, the time to reduce parasite density below 1% and parasite clearance were also collected. Patients were divided into two groups, namely those requiring and those not requiring intensive care (ICU and non-ICU, respectively). The ICU criteria for admission were: Glasgow Coma Scale <11, presence of multiple convulsions, respiratory distress with a PaO2/FiO2 Ratio <100, conditions requiring mechanical or non-invasive ventilation, circulatory collapse or shock and/or mean arterial pressure <70 mmHg requiring vasopressor drugs, all those conditions requiring close monitoring and/or prolonged sedation. For each patient, five prognostic malaria scores were calculated: the Malaria Score for Adults (MSA) [8], the Coma Acidosis Malaria score (CAM) [9], the Respiratory rate-based CAM score (R-CAM) [9], the Malaria Severity Score (MSS) [10] and the GCRBS (Glasgow coma scale, Creatinine, Respiratory rate, Bilirubin, Systolic blood pressure) Score [11]. Clinical severity at ICU admission was

also assessed using the Sequential Organ Failure Assessment (SOFA) score [12] and quick-SOFA (q-SOFA) score [13].

## Statistical analysis

Comparison of continuous data between 2 groups were analysed with the Student's t test or Mann-Whitney test, as appropriate. Fisher exact test was used to evaluate age, gender and WHO criteria distribution. Pearson's correlation and linear regression were used to evaluate the correlation between malarial and non-malarial scores. Data were expressed as median (interquartile range, IQR) or mean ± standard deviation (M±SD), as appropriate. Univariate and multivariate analyses were performed to analyse the factors independently associated to admission in ICU. The final multivariate model was selected with a forward stepwise variable selection procedure. In the multivariate analysis, a stepwise logistic regression model was fitted. A $p$ value of $<0.05$ was considered statistically significant. Statistical analyses were performed using GraphPad PRISM and Stata Corp. 2013. *Stata Statistical Software*: *Release 13*. College Station, TX: StataCorp LP.

# Results

## Study population

From 2007 to 2015, a total of 98 patients with severe malaria were admitted; 10 of them requiring IC (Table 1).

Patients requiring IC showed longer length of hospitalization and parasite clearance time compared to the other group. Notably, there were 5 cases of *P. vivax* severe malaria, including one requiring IC. Overall, the reasons for IC were cerebral malaria (7 cases), shock (6 cases) and acute respiratory distress syndrome (4 cases). The presence of 2 concurrent criteria for IC was observed in seven patients. All patients who required IC were treated with either intravenous artesunate (2.4 mg/kg at 0, 12, 24 and 48 hours, in 6 cases) or quinine (20 mg/kg loading dose followed by 10 mg/kg for a total of 48 hours, in 4 cases). Moreover, all ICU patients received antibiotic treatment for bacterial infections (3 central venous catheter-associated blood stream infections due to multidrug resistant *Acinetobacter baumannii*, methicillin-resistant *Staphylococcus aureus*, and *Candida albicans*, 6 ventilator associated pneumoniae, 1 catheter-associated urinary tract infection). In addition, eight patients needed mechanical ventilation for acute lung injury (6 cases) and acute respiratory distress syndrome (2 cases), 6 patients were treated with vasopressor drugs for shock and 2 patients with continuous venovenous hemodiafiltration for acute renal failure. At 28-day follow-up, no deaths or documented sequelae were reported.

## Malarial and non-malarial severity scores

The different malarial scores were estimated for all 98 severe malaria patients; those who required IC had higher scores than the others (Table 2).

The relation between SOFA, q-SOFA and malarial severity scores was evaluated and a positive linear regression was found between SOFA and GCRBS only (p = 0.01, $r^2$ = 0.58) (Fig 1).

## ICU versus non-ICU patients

Mean age, baseline parasite density, platelet count, haemoglobin and serum creatinine levels were similar between ICU and non-ICU patients; ICU patients had a slower parasite clearance (127.2±19 *vs* 84.7±4.6 hours, p = 0.006) and higher total bilirubin (4.8±1.2 *vs* 2±0.16 mg/dl, p<0.0001), aspartate aminotransferase (AST, 159.3±34 *vs* 66.5±7.6 U/L, p = 0.0003) (Fig 2A)

**Table 1. Severe malaria patients by Intensive Care need (%).**

| Variables | ICU | NON-ICU | p |
|---|---|---|---|
| | (n = 10) | (n = 88) | |
| Male | 8 (80) | 71 (80.7) | 0.421 |
| Age (years) (M±SD) | 45 ± 14 | 38 ± 13 | 0.142 |
| Country of origin | | | |
| - European | 6 (60) | 43 (49) | |
| - Immigrant | 4 (40) | 41 (46.5) | 0.223 |
| - Visitors from endemic country | 0 | 4 (4.5) | |
| Comorbidities | | | |
| - HIV | 1 (10) | 4 (4.5) | |
| - Chronic Viral Hepatitis | 1 (10) | 1 (1.1) | |
| - Arterial Hypertension | 1 (10) | 10 (11.4) | |
| - COPD | 0 | 4 (4.45) | 0.443 |
| - Splenectomy | 1 (10) | 0 | |
| - Cancer | 0 | 1 (1.1) | |
| - Hypothyroidism | 0 | 2 (2.3) | |
| Antimalarial chemoprophylaxis (n, %) | 0 | 9 (10.2) | 0.552 |
| Area of infection (n, %): | | | |
| West Africa | 10 (100) | 65 (73.9) | 0.944 |
| Purpose of travel (n, %) | | | |
| - VRFs | 2 (20) | 30 (34.1) | |
| - Tourism | 3 (30) | 25 (28.4) | 0.876 |
| - Business | 4 (40) | 13 (14.8) | |
| - Humanitarian aid | 1 (10) | 10 (11.4) | |
| - Other | 0 | 10 (11.4) | |
| *Plasmodium* (n, %) | | | |
| - *Falciparum* | 9 (90) | 83 (94.3) | |
| - *Vivax* | 1 (10) | 4 (4.5) | 0.554 |
| - Mixed Infection | 0 | 1 (1.1) | |
| Delay of diagnosis (days) (M±SD) | 4.5 ± 0.8 | 4.2 ± 0.5 | 0.851 |
| Delay of treatment (days) (M±SD) | 4.9 ± 0.7 | 4.8 ± 0.5 | 0.951 |
| Days of hospitalization (M±SD) | 25.6 ± 7.4 | 7.1 ± 0.5 | **0.0001** |
| Basal % parasitemia (Median, IQR) | 7 (2–7) | 5 (0.9–7) | 0.811 |
| Time to parasitaemia <1% (hours) (M±SD) | 67.2 ± 11.7 | 51.5 ± 3.0 | 0.122 |
| Time to parasite clearance (hours) (M±SD) | 127.2 ± 19.6 | 84.7 ± 4.6 | **0.006** |
| Antimalarial Treatment during Hospitalization | | | |
| - Artesunate (iv) | 6 (60) | 17 (19.3) | |
| - Quinine (iv) | 4 (40) | 17 (19.3) | |
| - Arthemeter (im) | 0 | 11 (12.5) | 0.434 |
| - Quinine (oral) | 0 | 17 (19.3) | |
| - Dihydroartemisinin/piperaquine | 0 | 24 (27.4) | |
| - Cloroquine | 0 | 2 (2.2) | |

ICU: Intensive Care Unit; im: intramuscular; iv: intravenous; M: mean; SD: Standard deviation; IQR: interquartile range, COPD: chronic obstructive pulmonary disease, VRFs:Visiting Relatives and Friends.

and alanine aminotransferase (ALT, 102.9±19.3 *vs* 61.3±7.3 U/L, p = 0.07). All ICU patients had more than 3 WHO criteria for severe malaria while the other patients met only one criterion (3.8±0.2 vs 1.3±0.11, p<0.001) (Fig 2B): specifically, the presence of neurological

**Table 2. Malarial score: ICU *vs* non-ICU.**

| Malarial Scores (mean) | ICU | NON-ICU | *p* |
|---|---|---|---|
| MSA | 5.4 | 0.9 | *<0.001* |
| CAM | 1.6 | 0.4 | *0.008* |
| R-CAM | 2.1 | 0.3 | *0.002* |
| MSS | 11.9 | 1 | *<0.001* |
| GCRBS | 2.9 | 0.3 | *<0.001* |

ICU: Intensive Care Unit, MSA: Malaria Score for Adults, CAM: Coma Acidosis Malaria score, R-CAM: Respiratory rate-based CAM score, MSS: Malaria Severity Score, GCRBS Score: Glasgow coma scale, Creatinine, Respiratory rate, Bilirubin, Systolic blood pressure.

(p<0.001), liver (p = 0.03) or respiratory involvement (p<0.001) as single WHO criterion of severe malaria was associated to IC need.

Univariate analysis showed that the number of WHO criteria (p<0.001), time to parasite clearance (p = 0.012), levels of serum AST (p = 0.004) and bilirubin (p = 0.003) at admission were associated with IC. Multivariate analysis showed that the number of WHO criteria (Odd Ratio [OR] = 2.58; confidence interval [CI] = 1.36–4.91; p = 0.004) and AST value (OR = 1.01; CI = 1.00–1.02; p = 0.037) were independently associated with IC (Table 3).

## Discussion

Malaria remains a substantial problem in non-endemic countries. In this series of patients recruited over a 8-year period, 10 patients required IC, with a 4-day median delay of malaria diagnosis; all of them had been infected in West Africa and none of them received anti-malaria chemoprophylaxis. In a non-endemic country, any imported severe malaria case is a medical emergency, often complicated by a delayed access to care, diagnosis and initiation of specific parenteral therapy. The capacity of the different national health systems for reporting malaria cases varies substantially and depends on the capacity for a reliable diagnosis [14]. Unspecific febrile symptoms and low quality microscopy reading in non-specialised laboratories can

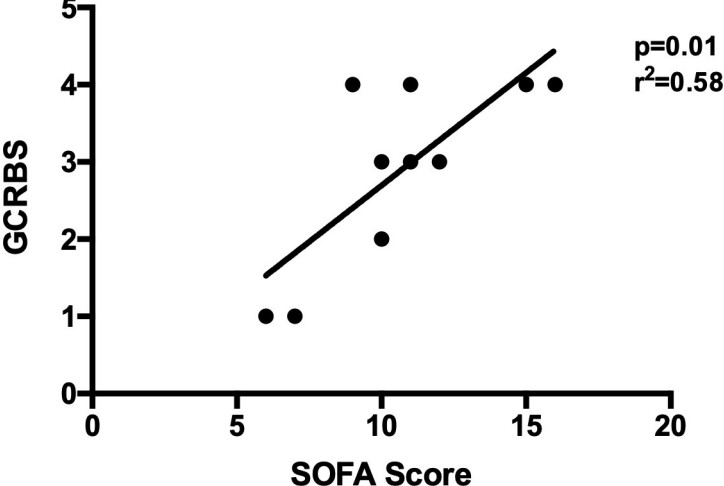

**Fig 1. Malarial and non-malarial severity scores.** A positive linear regression was found between Sequential Organ Failure Assessment score (SOFA) and GCRBS Score (Glasgow coma scale, Creatinine, Respiratory rate, Bilirubin, Systolic blood pressure).

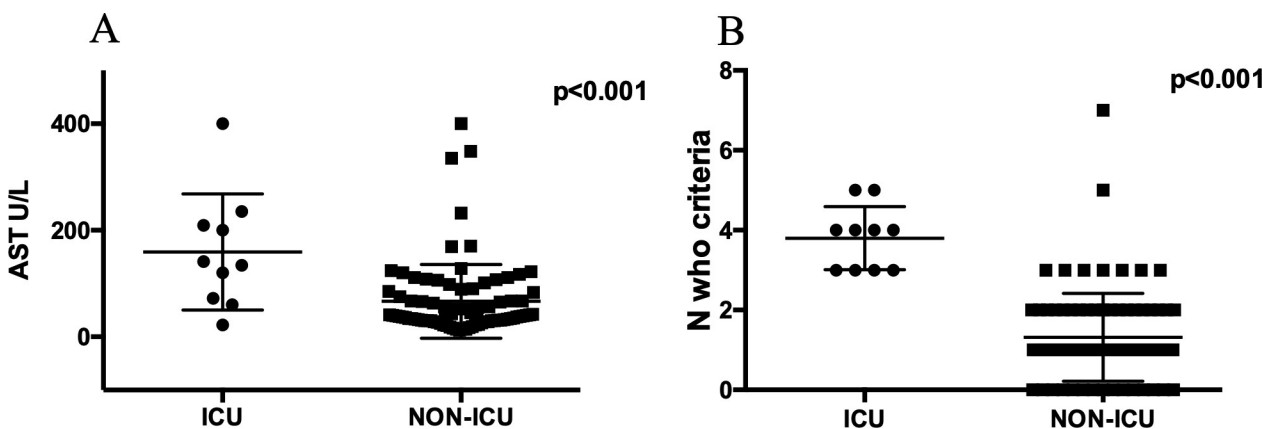

**Fig 2. Aspartate aminotransferase levels (A) and N_WHO criteria (B) between ICU and non-ICU patients.** Intensive Care Unit (ICU) group showed a significantly higher aspartate aminotransferase (AST) value (A) and higher number of WHO criteria (B) compared to non-ICU group.

delay diagnosis [15]. In this series of patients, *P. falciparum* was the main causing species although there were some *P. vivax* cases, with one of them requiring IC. There have been increasing reports of severe malaria caused by *P. vivax* [2].

Severe malaria may rapidly evolve to an unfavourable prognosis requiring IC with a case-fatality rate between 5 and 10%. Therefore, identifying severe malaria patients at higher risk of death is of paramount importance. However, not all WHO criteria have the same predictive value for the early identification of patients with unfavourable prognosis. In the 400-patient cohort study conducted in France by Bruneel et al., three baseline variables independently predicted death: older age, coma and high parasite density [16]. In our study, five prognostic malaria scores were calculated for each ICU patient (MSA, CAM, R-CAM, MSS, GCRBS score) and clinical severity at ICU admission was assessed using both SOFA and q-SOFA scores. In this study, most ICU patients had a medium-high SOFA score with a high SOFA-related mortality predictive value. In addition, there was a positive correlation between SOFA, a non-specific malaria score, and GCRBS, a specific malaria score. An early assessment of the severity status of the patient by specific score is required at admission to rapidly decide whether the patient needs to be admitted in ICU. Applying both malaria-specific (GCRBS)

**Table 3. Logistic regression model to evaluate risk factors associated with IC.**

| | Univariate | | Multivariate* | |
|---|---|---|---|---|
| | **OR (95% CI)** | ***p*** | **OR (95% CI)** | ***p*** |
| N_WHO criteria | 4.00 (1.96–8.16) | *<**0.001*** | 2.58 (1.36–4.92) | ***0.004*** |
| Basal parassitemia | 0.97 (0.75–1.24) | *0.816* | | |
| Time parasitemia clearance | 1.01 (1.00–1.03) | ***0.012*** | | |
| Platelet count | 1.00 (0.99–1.00) | *0.204* | | |
| Haemoglobin | 0.83 (0.65–1.07) | *0.166* | | |
| Creatinine | 1.72 (0.39–7.51) | *0.466* | | |
| Bilirubine | 1.74 (0.12–2.52) | ***0.003*** | 1.52 (0.93–2.47) | *0.089* |
| AST | 1.10 (1.00–1.01) | ***0.004*** | 1.01 (1.00–1.02) | ***0.037*** |
| ALT | 1.0 (0.99–1.01) | *0.092* | | |

OR: Odds Ratio; CI: confident interval;

*with forward stepwise variable selection procedure, WHO: World Health Organization, AST: aspartate aminotransferase, ALT: alanine aminotransferase.

and general (SOFA) scores to severe malaria patients could be the best approach to assess the need for IC.

The findings of this study have few limitations. First, this is a retrospective and cross-sectional study conducted in a single centre. Second, the small number of enrolled patients, especially those requiring ICU admission, could affect the generalization of our results to a wider population. However, our results are consistent with data from previous studies. In a recent review on the clinical management of imported severe malaria in adults, the authors recommended an assessment by ICU staff for all patients with malaria even if presenting one severity criterion, only [17]. Moreover, patients with uncomplicated malaria but at risk of rapidly evolving towards severe malaria should be under continuous monitoring; a >2 q-SOFA score should be considered as an indication for ICU admission [17]. In our case series, patients requiring IC had a higher number of WHO criteria than the other group, and cerebral malaria, liver and respiratory failures were the most frequent observed complications. Indeed, at multivariate analysis, the number of WHO criteria and the AST level were independently associated with ICU admission. A prompt intravenous antimalarial regimen with adequate supportive therapy warranted a favourable outcome: the WHO guidelines for the treatment of severe malaria recommend the use of intravenous artesunate for at least the first 24 hours [2]. Such recommendation is based on the results of two large trials carried out in malaria-endemic countries that showed artesunate superiority to quinine in terms of efficacy (rapid decay of parasitaemia), safety (no major adverse events) and better survival [18,19]. All our patients requiring IC were treated with either quinine (4 patients before 2010 when parenteral artesunate was unavailable) or artesunate. All patients recovered without any complications or sequelae.

## Conclusions

Considering that severe malaria may rapidly evolve to an unfavourable prognosis, a prompt and accurate evaluation for ICU admission is needed. According to our results, the assessment of both malarial and non-malarial severity scores may contribute to the proper management of severe malaria. Specifically, the number of WHO criteria and AST plasma level can predict the need of intensive care.

## Supporting information

**S1 Table. Strobe checklist.**
(DOCX)

**S1 File. Dataset.**
(XLSX)

## Acknowledgments

We acknowledge the Spallanzani Group for Malaria: Tommaso Ascoli Bartoli, Federica Calò, Antonino Di Caro, Gabriele Fabbri, Luisa Marchioni, Assunta Navarra, Carla Nisii, Maria Grazia Paglia, Claudia Palazzolo, Paola Scognamiglio, Francesco Vairo.

## Author Contributions

**Conceptualization:** Alessandra D'Abramo, Umberto D'Alessandro, Emanuele Nicastri.

**Data curation:** Alessandra D'Abramo, Luciana Lepore, Saba Gebremeskel Tekle, Angela Cor-
polongo, Laura Scorzolini, Nazario Bevilacqua, Maria Letizia Giancola, Umberto
D'Alessandro.

**Formal analysis:** Marco Iannetta, Angela Corpolongo.

**Funding acquisition:** Emanuele Nicastri.

**Investigation:** Saba Gebremeskel Tekle, Laura Scorzolini, Nazario Bevilacqua, Andrea Mari-
ano, Maria Letizia Giancola, Antonella Vulcano, Micaela Maritti, Mario Antonini, Ema-
nuele Nicastri.

**Methodology:** Marco Iannetta, Andrea Mariano, Alessandro Agresta.

**Software:** Alessandro Agresta.

**Supervision:** Alessandra D'Abramo.

**Writing – original draft:** Alessandra D'Abramo, Luciana Lepore.

**Writing – review & editing:** Alessandra D'Abramo, Mario Antonini, Umberto D'Alessandro,
Emanuele Nicastri.

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
