## [Decision Letter · Decision Letter 0]

4 Sep 2019

PONE-D-19-16792

Imported severe malaria and risk factors for intensive care: a single-centre retrospective analysis.

PLOS ONE

Dear Dr Iannetta,

Thank you for submitting your manuscript to PLoS ONE. After careful consideration, we felt that your manuscript requires substantial revision, following which it can possibly be reconsidered, thus governing the decision of a “major revision”. Regarding the study design, there are significant concerns raised by reviewers. According to Reviewer # 1, the criteria for IC admission is not clear and whether or not patients presented dual pathologies.  Reviewer #2 recommend to use the STROBE checklist to improve the reporting quality of results. Of further relevance, both reviewers complain that conclusion must translate the results, and authors needs to adjust that. Finally, the low number of IC patients may compromise the MS’s conclusion, and authors should include study limitation. For your guidance, a copy of the reviewers' comments was included below.   We therefore invite you to submit a revised version of the manuscript paying close attention to the specific points raised by both reviewers. 

We would appreciate receiving your revised manuscript by September 30 . To enhance the reproducibility of your results, we recommend that if applicable you deposit your laboratory protocols in protocols.io, where a protocol can be assigned its own identifier (DOI) such that it can be cited independently in the future. For instructions see: http://journals.plos.org/plosone/s/submission-guidelines#loc-laboratory-protocols

We look forward to receiving your revised manuscript.

Kind regards,

Luzia Helena Carvalho, Ph.D.

Academic Editor

PLOS ONE

Journal Requirements:

Reviewers' comments:

Reviewer's Responses to Questions

**Comments to the Author**

1. Is the manuscript technically sound, and do the data support the conclusions?

Reviewer #1: Partly

Reviewer #2: Yes

2. Has the statistical analysis been performed appropriately and rigorously? 

Reviewer #1: Yes

Reviewer #2: Yes

3. Have the authors made all data underlying the findings in their manuscript fully available?

Reviewer #1: No

Reviewer #2: No

4. Is the manuscript presented in an intelligible fashion and written in standard English?

Reviewer #1: Yes

Reviewer #2: Yes

5. Review Comments to the Author

Reviewer #1: This a retrospective single-centre review of imported severe malaria. I have the following comments:

1) What were the pre-existing admission criteria for admission to IC?

2) A total of 10 cases needing IC are reported although the reason for admission were cerebral malaria (7), shock (6) a nd ARDS (4) which totals 17, were there any patients that had dual pathologies?

3) ICU vs non-ICU patients had slower parasitemia but did the ICU patients have higher parasitemia to start off with which could explain the slower clearance?

4) 6 patients were reported as requiring vasoplressor drugs for shock but there is no mention of fluid resuscitation used. This may relevant particualry in light of the FEAST study.

5) There were no deaths or sequelae reported yet the conclusions state 'a multidisciplinary approach might be encouraged... in order to improve outcome', what outcome are they trying to improve? Furthermore what and whom should be part of a multi-disciplinary approach'?

Reviewer #2: This is a simple, concise cross-sectional study of medical record data from patients with severe malaria to assess predictive risk factors associated to the need of intensive care (IC) at a referral hospital in Italy. Authors used five prognostic malaria scores to analyse the need of IC and applyed malaria-specific and general scores to identify severe malaria patients.

The manuscript is well written, clear and intelligible. However, authors should make some changes to the manuscript in order to better interpret and conclude the results.

1) Authors are recommended to use the STROBE checklist to improve the reporting quality of results.

2) Some limitations of the study should be discussed, such as the low number of patients requiring intensive care, especially if the species was P. vivax. In addition, the cross-sectional design of the study precludes a more conclusive inference from the study results.

3) The conclusion of the study should be presented more clearly and objectively. In the Summary, for example, the conclusion presented does not translate the results found in the study.

6. PLOS authors have the option to publish the peer review history of their article (what does this mean?). If published, this will include your full peer review and any attached files.

Reviewer #1: No

Reviewer #2: No

---

## [Author Response · Author response to Decision Letter 0]

27 Sep 2019

REVIEWER #1

This a retrospective single-centre review of imported severe malaria. I have the following comments:

1) What were the pre-existing admission criteria for admission to IC?

A: The admission criteria adopted for IC admission were:

- Glasgow Coma Scale <11;

- Presence of multiple convulsions;

- Respiratory distress with PaO2/FiO2 Ratio <100);

- Conditions requiring mechanical ventilation and/or non-invasive ventilation;

- Circulatory collapse or shock and/or MAP <70 mmHg, requiring vasopressor drugs;

- Conditions requiring close monitoring and/or prolonged sedation.

We included the IC admission criteria in the manuscript.

2) A total of 10 cases needing IC are reported although the reason for admission were cerebral malaria (7), shock (6) and ARDS (4) which totals 17, were there any patients that had dual pathologies?

A: In our study, 7 patients presented more than one clinical picture at the moment of the admission in the IC, therefore the number of the events (17) is greater than the number of patient (10). A sentence was added in the text, accordingly. 

3) ICU vs non-ICU patients had slower parasitemia but did the ICU patients have higher parasitemia to start off with which could explain the slower clearance?

A: As showed in Table 1, there is no difference in basal parasitemia between ICU vs non-ICU (p= 0.811). Moreover, the data is confirmed at the univariate analysis.

4) 6 patients were reported as requiring vasopressor drugs for shock but there is no mention of fluid resuscitation used. This may relevant particuarly in light of the FEAST study.

A: According to the protocol of our Institute, the use of vasopressor drugs is secondary to the fluid resuscitation failure.

5) There were no deaths or sequelae reported yet the conclusions state 'a multidisciplinary approach might be encouraged... in order to improve outcome', what outcome are they trying to improve? Furthermore what and whom should be part of a multi-disciplinary approach'?

A: As suggested by the reviewer, we modified the conclusion of the manuscript.

REVIEWER #2

This is a simple, concise cross-sectional study of medical record data from patients with severe malaria to assess predictive risk factors associated to the need of intensive care (IC) at a referral hospital in Italy. Authors used five prognostic malaria scores to analyse the need of IC and applyed malaria-specific and general scores to identify severe malaria patients. The manuscript is well written, clear and intelligible. However, authors should make some changes to the manuscript in order to better interpret and conclude the results.

1) Authors are recommended to use the STROBE checklist to improve the reporting quality of results.

A: According to reviewer’s suggestion we modified the methods section using the STROBE checklist, which will be uploaded as a supplementary file.

2) Some limitations of the study should be discussed, such as the low number of patients requiring intensive care, especially if the species was P. vivax. In addition, the cross-sectional design of the study precludes a more conclusive inference from the study results.

A: We thank the reviewer for the suggestions. The text has been modified accordingly, introducing sentences showing some limitations of the study. 

3) The conclusion of the study should be presented more clearly and objectively. In the Summary, for example, the conclusion presented does not translate the results found in the study.

A: The conclusion has been modified accordingly, introducing sentences that reflect the results showed in the text.

---

## [Decision Letter · Decision Letter 1]

7 Oct 2019

PONE-D-19-16792R1

Imported severe malaria and risk factors for intensive care: a single-centre retrospective analysis.

PLOS ONE

Dear Dr Iannetta,

Thank you for submitting your manuscript for review to PLoS ONE. After careful consideration, we feel that your manuscript will likely be suitable for publication if it is revised to address major points raised now by the reviewers. Specifically, the authors should clarify a couple of topics related to study design, including why they requested written informed consent from patients’ next of kin, since all patients are adults.   A significant number of grammatical errors remained in the revised version of MS. Thus, the language needs to be properly adjusted otherwise it might compromise the publication. Finally, the reviewers complain about Data Availability that is not entirely accessible as stated by the authors. 

We would appreciate receiving your revised manuscript by October 30. To enhance the reproducibility of your results, we recommend that if applicable you deposit your laboratory protocols in protocols.io, where a protocol can be assigned its own identifier (DOI) such that it can be cited independently in the future. For instructions see: http://journals.plos.org/plosone/s/submission-guidelines#loc-laboratory-protocols

We look forward to receiving your revised manuscript.

Kind regards,

Luzia Helena Carvalho, Ph.D.

Academic Editor

PLOS ONE

Reviewers' comments:

Reviewer's Responses to Questions

**Comments to the Author**

1. If the authors have adequately addressed your comments raised in a previous round of review and you feel that this manuscript is now acceptable for publication, you may indicate that here to bypass the “Comments to the Author” section, enter your conflict of interest statement in the “Confidential to Editor” section, and submit your "Accept" recommendation.

Reviewer #2: All comments have been addressed

Reviewer #3: All comments have been addressed

2. Is the manuscript technically sound, and do the data support the conclusions?

Reviewer #2: Yes

Reviewer #3: Yes

3. Has the statistical analysis been performed appropriately and rigorously? 

Reviewer #2: Yes

Reviewer #3: Yes

4. Have the authors made all data underlying the findings in their manuscript fully available?

Reviewer #2: No

Reviewer #3: Yes

5. Is the manuscript presented in an intelligible fashion and written in standard English?

Reviewer #2: (No Response)

Reviewer #3: Yes

6. Review Comments to the Author

Reviewer #2: Although the authors stated that the study data is entirely available within the manuscript and in supplementary files, I have not had access to it completely.

Reviewer #3: This is a cross sectional and retrospective study where clinical records for patients who were admitted with severe malaria from 2007 to 2015 were reviewed and analyzed by the authors. In this revised version of the manuscript, the authors have satisfactorily addressed comments that were raised by two reviewers. The manuscript is technically sound with the data used supporting the conclusion of the paper. All statistical analyses have been done appropriately and rigorously. Nevertheless I have the following concerns with the revised manuscript which the authors need to also look into:

1. The presentation of the manuscript is intelligible and uses Standard English though it will benefit from some language usage proofreading.

2. The authors need to clearly distinguish between actual recruitment of the patients and use of records for this research study. An example is where they write that they obtained written informed consent from the patients or their next of kin before enrollment into the study without detailing how they did it considering that by the time of this research the patients were not in hospital.

3. Further the authors write that they obtained written informed consent from patients’ next of kin, since all patients are adults aged above 18 years why was consent obtained from next of kin?

4. Authors should spell out all abbreviations the first time they are used in the write up – examples are INMI, h, P. vivax etc.

7. PLOS authors have the option to publish the peer review history of their article (what does this mean?). If published, this will include your full peer review and any attached files.

Reviewer #2: No

Reviewer #3: Yes: Peter Makaula

---

## [Author Response · Author response to Decision Letter 1]

25 Oct 2019

REVIEWER #2 

1. Although the authors stated that the study data is entirely available within the manuscript and in supplementary files, I have not had access to it completely.

A: In the previous revision, we have already uploaded the complete database as Supplementary file (page 23, download link). We will upload again the entire database as a supplementary file. If this is not enough, please let us know what we can do further.

REVIEWER #3: 

This is a cross sectional and retrospective study where clinical records for patients who were admitted with severe malaria from 2007 to 2015 were reviewed and analyzed by the authors. In this revised version of the manuscript, the authors have satisfactorily addressed comments that were raised by two reviewers. The manuscript is technically sound with the data used supporting the conclusion of the paper. All statistical analyses have been done appropriately and rigorously. Nevertheless I have the following concerns with the revised manuscript which the authors need to also look into:

1. The presentation of the manuscript is intelligible and uses Standard English though it will benefit from some language usage proofreading.

A: As suggested by the reviewer, the manuscript was widely revised for language editing.

2. The authors need to clearly distinguish between actual recruitment of the patients and use of records for this research study. An example is where they write that they obtained written informed consent from the patients or their next of kin before enrollment into the study without detailing how they did it considering that by the time of this research the patients were not in hospital.

A: As suggested by the reviewer, we modified the text. According to the protocol in use in our Institute, each patient signs an informed written consent at the time of admission for research purposes and epidemiological investigations on infectious and tropical diseases. 

3. Further the authors write that they obtained written informed consent from patients’ next of kin, since all patients are adults aged above 18 years why was consent obtained from next of kin?

A: As suggested by the reviewer we modified the text. In case of person unable (e.g. coma state) to give consent, a relative/next of kin was asked to provide it.

4. Authors should spell out all abbreviations the first time they are used in the write up – examples are INMI, h, P. vivax etc.

A: As suggested by the reviewer, we modified the text.

---

## [Editor Report · Decision Letter 2]

30 Oct 2019

Imported severe malaria and risk factors for intensive care: a single-centre retrospective analysis.

PONE-D-19-16792R2

Dear Dr. Iannetta,

We are pleased to inform you that your manuscript has been judged scientifically suitable for publication and will be formally accepted for publication once it complies with all outstanding technical requirements.

With kind regards,

Luzia Helena Carvalho, Ph.D.

Academic Editor

PLOS ONE
---

## [Editor Report · Acceptance letter]

5 Nov 2019

PONE-D-19-16792R2 

Imported severe malaria and risk factors for intensive care: a single-centre retrospective analysis. 

Dear Dr. Iannetta:

I am pleased to inform you that your manuscript has been deemed suitable for publication in PLOS ONE. Congratulations! Your manuscript is now with our production department. 

With kind regards,

on behalf of

Dr. Luzia Helena Carvalho 

Academic Editor

PLOS ONE